# Comprehensive Analysis of the lncRNA–miRNA–mRNA Regulatory Network for Intramuscular Fat in Pigs

**DOI:** 10.3390/genes14010168

**Published:** 2023-01-07

**Authors:** Yanhui Zhao, Shaokang Chen, Jiani Yuan, Yumei Shi, Yan Wang, Yufei Xi, Xiaolong Qi, Yong Guo, Xihui Sheng, Jianfeng Liu, Lei Zhou, Chuduan Wang, Kai Xing

**Affiliations:** 1Animal Science and Technology College, Beijing University of Agriculture, Beijing 102206, China; 2Beijing Animal Husbandry Station, Beijing 100101, China; 3College of Animal Science and Technology, China Agricultural University, Beijing 100193, China

**Keywords:** intramuscular fat, fatty acid, regulatory network

## Abstract

Intramuscular fat (IMF) is an essential trait closely related to meat quality. The IMF trait is a complex quantitative trait that is regulated by multiple genes. In order to better understand the process of IMF and explore the key factors affecting IMF deposition, we identified differentially expressed mRNA, miRNA, and lncRNA in the *longissimus dorsi* muscle (LD) between Songliao Black (SL) pigs and Landrace pigs. We obtained 606 differentially expressed genes (DEGs), 55 differentially expressed miRNAs (DEMs), and 30 differentially expressed lncRNAs (DELs) between the SL pig and Landrace pig. Enrichment results from GO and KEGG indicate that DEGs are involved in fatty acid metabolism and some pathways related to glycogen synthesis. We constructed an lncRNA–miRNA–mRNA interaction network with 18 DELs, 11 DEMs, and 42 DEGs. Finally, the research suggests that *ARID5B*, *CPT1B*, *ACSL1*, *LPIN1*, *HSP90AA1*, *IRS1*, *IRS2*, *PIK3CA*, *PIK3CB*, and *PLIN2* may be the key genes affecting IMF deposition. The LncRNAs MSTRG.19948.1, MSTRG.13120.1, MSTRG.20210.1, and MSTRG.10023.1, and the miRNAs ssc-miRNA-429 and ssc-miRNA-7-1, may play a regulatory role in IMF deposition through their respective target genes. Our research provides a reference for further understanding the regulatory mechanism of IMF.

## 1. Introduction

Pork is the main source of human protein intake. In recent decades, the intramuscular fat (IMF) content of modern pig breeds has decreased due to the progress of breeding [1]. The content of IMF is closely related to the quality of pork, which can affect the tenderness and water-holding capacity of meat [2,3,4]. The flavor of pork is improved when the intramuscular fat content ranges between 2.5% and 3.5% [2]. To better meet the needs of consumers and the market, research on the factors affecting IMF deposition has become a popular area.

IMF is composed of phospholipids, triglycerides, and cholesterol. At the metabolic level, the IMF content depends on the balance between the uptake, synthesis, and degradation of triacylglycerol. At the species level, IMF deposition is the result of the balance between fat intake, liver de novo synthesis, and lipoprotein lipase (LPL) [5]. The IMF deposition process is complex and regulated by multiple genes, including mRNAs, miRNAs, and lncRNAs. Previous studies have shown that lncRNA participates in the regulatory network of adipogenesis in various ways and affects the fat deposition content. For example, the knockdown of lncimf4 can promote the proliferation of intramuscular adipocytes [6], and lnc_000414 can inhibit the proliferation of porcine intramuscular adipocytes [7]. Similarly, miRNA also plays a significant role in adipogenesis. Studies have shown that miR-17, miR-21, and miR-143 can promote preadipocyte differentiation in pigs, while miR-145 and miR-429 can inhibit preadipocyte differentiation [8,9,10]. MiRNA can play a role as the target gene of lncRNA and miRNA, and lncRNA can also regulate miRNA [11]. However, there have been few studies on the joint analysis of lncRNA, miRNA, and mRNA to explore the impact on IMF deposition.

Landrace pigs are typical lean pigs, while Songliao black (SL) pigs have greater fat deposition ability, so it is an ideal to study fat deposition [12]. To better understand the factors affecting IMF deposition, we performed a comprehensive analysis of the lncRNA–miRNA–mRNA regulatory network in the intramuscular fat of SL and Landrace pigs.

## 2. Materials and Methods

### 2.1. Experimental Animals and Sample Collection

In this study, gilt SL (*n* = 6, IMF, 3.60 ± 1.23%) and Landrace pigs (*n* = 6, IMF, 1.36 ± 0.16%) were selected, and both came from the Tianjin Ninghe original pig farm. The pigs used in this study were all raised indoors, and the hog houses were cleaned and disinfected weekly, with no additional environmental controls. All pigs in the study were provided ad libitum access to water and feed (feed composition: corn 63%, soybean meal 8.82%, wheat bran 23.5%, soybean oil 1.11%, calcium bicarbonate 1.22%, calcium carbonate 0.95%, salt 0.2%, premix 1.2%), and in good health. The pigs of both breeds were slaughtered when raised to approximately 100 kg under the same feeding conditions, and pain was minimized during slaughter. The average slaughter age of Landrace was 176 and that of SL was 190. After slaughter, the *longissimus dorsi* muscle (LD) tissue from the third to fourth ribs was quickly collected and placed in liquid nitrogen for subsequent use.

### 2.2. RNA Extraction and Sequencing

Total RNA was isolated from the LD of 12 pigs using a Trizol reagent. The concentration and quality of RNA were verified by measuring ratio of absorbance at 260 nm and 280 nm (A260/A280) using Smart Spec Plus (bio rad, Hercules, CA, USA). A total of 1% agarose gel electrophoresis was used to detect the quality and integrity of RNA [13]. Concerning the instructions of the reverse transcription kit (Invitrogen, Carlsbad, CA, USA), the cDNA library was prepared for the extracted RNA. After purification and enrichment, all libraries were sequenced by Illumina HiSeq 2000 (Illumina, San Diego, CA, USA). The length of the sequence for mRNA and lncRNA was 150 bp and the depth of the sequence was 3X. The miRNA sequence length was 80 bp.

### 2.3. Mapping and Assembly of Sequenced RNA Data

Raw data were filtered using fastp (v0.12.4) [14] to remove reads with N and base mass value (Q) below 20. The quality of sequencing data was evaluated using FastQC (v0.11.9) [15]. The porcine reference genome sequence (version: Sus scrofa v. 11.1) and genome annotation file (version: Sus_scrofa.Sscrofa11.1.108) were downloaded from the Ensembl database (https://asia.ensembl.org/Sus_scrofa/Info/Index (accessed on 15 March 2021)). The index was constructed using Hisat2 (v2.2.1) [16], and the clean reads were aligned with the reference genome. Samtools (v1.15) [17] was used to sort Sam files and convert them to BAM files. Then, the mRNA was quantified by HTseq (v2.0.1) [18] software, and the transcripts of each sample were assembled using StringTie (v2.2.1) [19] to obtain 12 gene transfer format (GTF) files. Finally, the GTF files of 12 samples were combined into a non-redundant GTF file by using the merge function of the StringTie software package [20]. MiRNA precursors and mature files were downloaded from the miRBase (v22.1) [21] database, and then mirdeep2 (v0.1.3) [22] was used for mapping and expression calculation.

### 2.4. Pipeline for lncRNA Identification

We used the following steps to identify lncRNAs from non-redundant GTF files, mainly including (Figure 1a): (1) Transcripts with class codes = “i, u, x” were retained. (2) Transcripts with exon number ≥2 and sequence length >200 were retained.(3) Five tools were used to identify lncRNAs, four of which were coding ability prediction tools, including Coding Potential Calculator 2 (CPC2) [23], Coding-Non-Coding-Index (CNCI) [24], Coding Potential Assessing Tool (CPAT) [25] and predictor of long non-coding RNAs and messenger RNAs based on an improved k-mer scheme (PLEK) [26]. We used the Hmmer3 [27] tool to determine whether the transcripts retained by the four tools had a significant hit rate in the Pfam database (E-value < 1 × 10^−5^), and finally discarded the transcript containing any known protein-coding domain. (4) We used fragments per kilobase of transcript per million mapped reads (FPKM) ≥ 0.1 as the standard to ensure expression, and lncRNAs meeting this condition were retained in at least one sample.

### 2.5. Screening of Differentially Expressed RNA

After obtaining the expression amounts of the three RNAs, we further screened the differentially expressed RNAs in the SL and Landrace pigs’ LD by using the edgeR (v3.34.1) [28] software package. We used the false discovery rate (FDR) < 0.05 and |fold change| ≥ 2 as criteria to identify differentially expressed RNA. The R software package pheatmap was used to draw the cluster heat map of three differentially expressed RNAs.

### 2.6. Gene Ontology and Pathway Enrichment Analysis

To explore the function of differentially expressed genes (DEGs), we used DAVID [29] to analyze the gene ontology of DEGs and used KOBAS 3.0 [30] to analyze the enrichment of the KEGG pathway. It was considered that GO and KEGG pathways with *p*-values < 0.05 were significantly enriched pathways.

### 2.7. Target Gene Prediction and Functional Analysis

We used miRDB [31], TargetScan [32], and Starbase [33], three target gene prediction websites, to predict the differentially expressed miRNA (DEMs) target genes, and intersections with DEGs. We used BEDTools (v2.30.0) [34] to search for 100 kb adjacent genes upstream and downstream of differentially expressed lncRNAs (DELs) and considered it as the target gene of cis-regulation of DELs. At the same time, if the absolute value of the Spearman correlation coefficient between the expression levels of lncRNA and mRNA exceeded 0.8 and *p* < 0.05, it was considered that there was a targeted relationship between them. Then, the target genes of DELs and DEGs were intersected. In addition, we also used miRDB to predict the targeted miRNA of DELs and intersected with DEMs.

We combined the target genes of DELs and DEMs, analyzed the KEGG pathway enrichment, and then imported it into the string [35] database to construct the protein–protein interaction (PPI) network, and analyzed the genes with the top 10 connectivity using Cytoscape [36]. After that, we extracted the intersection between the potential differentially expressed target genes (DEPTGs) of each DEL and DEM and removed another mRNA. Finally, we brought the three target relationships of DEL to DEM, DEL to intersection DEPTG, and DEM to intersection DEPTG into the cytoscape tool to build an lncRNA–miRNA–mRNA interaction network, which was convenient for us to better understand the regulatory relationship between the three.

## 3. Results

### 3.1. Overview of RNA Sequencing Data

The raw mRNA and lncRNA sequencing data totaled 96 GB, and 256 million raw reads were obtained. The raw miRNA sequencing data totaled 20.1 GB, and 144 million raw reads were obtained. After mapping the clean reads to the pig genome, we found that the mapping rate of all samples was more than 85% (Appendix A).

### 3.2. Identification and Characterization of lncRNA

After using StringTie and StringTie merge, we obtained a GTF file with 1,254,032 transcripts. After that, we identified lncRNA in strict accordance with the process in Figure 1a, and finally predicted 4190 lncRNAs (Figure 1a). Most lncRNAs had lengths between 201 and 400 nt. The number of lncRNAs with two exons was also the largest.

### 3.3. Differential Expression RNA Analysis

By analyzing the differential expression of three RNAs between Landrace pigs and SL pigs, we found 606 differentially expressed mRNAs, of which 384 were significantly up-regulated and 222 were significantly down-regulated in the LD of SL pigs (Figure 2a, Appendix A). We also found 55 DEMs and 30 DELs in both pig breeds. A total of 19 miRNAs and 21 lncRNAs were significantly up-regulated in SL pigs, and 36 miRNAs and 9 lncRNAs were significantly down-regulated (Figure 2b,c, Appendix A). Next, we performed a cluster analysis on the three differentially expressed RNAs, and the heatmaps showed that the expression patterns of mRNA and lncRNA were consistent across groups, but not within groups, and the expression patterns of miRNA were not consistent across groups. (Figure 2d–f).

### 3.4. GO and KEGG Functional Enrichment Analysis of DEGs

We carried out GO and KEGG enrichment analysis on 606 DEGs, and the results are shown in Figure 3. In the enrichment results of GO, we found that there was significant enrichment in the biological processes related to fat metabolism and the deposition processes, such as the fatty acid metabolic process, and long-chain fatty acid transport in Figure 3a, and some pathways related to glycogen syntheses, such as the positive regulation of glycogen biological process. Meanwhile, the AMPK signaling pathway, fatty acid metabolism, and insulin resistance in KEGG enrichment results were considered to be important pathways related to fat deposition (Figure 3b).

### 3.5. Target Gene Prediction of Differentially Expressed miRNA and lncRNA

We used TargetScan, Starbase, and miRDB to predict the target genes of 54 DEMs, and obtained 17,786, 10,978, and 13,398 target genes, respectively (Appendix A). Then, we compared the three targeting results with DEGs and found 363 coincident genes (Figure 4a). Thirty DELs targeted 104 miRNAs, but only 16 were DEMs (Appendix A).

LncRNA can regulate genes through cis and trans actions. In our study, two methods were used to predict the target genes of lncRNA. Firstly, we used BEDTools to search the adjacent coding protein genes of DELs (within 100 kb upstream and downstream) and obtained 157 coding protein genes, but only 10 coincided with DEGs. Then, we predicted the target genes of lncRNA trans-regulation and found that 445 DEPTGs corresponded to 28 DELs, including 7 cis DEPTGs (Table 1). Additionally, the number of DEPTGs corresponding to different lncRNAs was very different. For example, lncRNA MSTRG.10885.1 has 196 DEPTGs, MSTRG.4262.2 has 108 DEPTGs, MSTRG.15373.1 has 57 DEPTGs, and MSTRG.7556.1 has only one DEPTG (Appendix A). Nine of the ten cis target genes of lncRNA coincided with the trans target genes. In addition, there were 285 coincident PTGs between lncRNA and miRNA (Figure 4b). We combined the target genes of lncRNA and miRNA to obtain 523 DEPTGs, which were then analyzed for KEGG pathway enrichment. The results indicate that DEPTGs were mainly enriched in insulin-related pathways, adipocyte-related pathways, and fatty-acid-related pathways (Table 2).

### 3.6. Analysis of PPI Network and lncRNA-miRNA-mRNA Regulatory Network

We imported 523 DEPTGs into the string database, constructed the PPI network, and then calculated the top ten genes with connectivity by using Cytoscape software, mainly including phosphatidylinositol-4,5-bisphosphonate 3-kinase catalytic subunit α (*PIK3CA*), phosphatidylinositol-4,5-bisphosphonate 3-kinase catalytic subunit β (*PIK3CB*), heat shock protein 90 α family class a member 1 (*HSP90AA1*), Erb-B2 receptor tyrosine kinase 2 (*ERBB2*) and other genes (Figure 5a). After that, we used Cytoscape software to construct the targeted regulatory network between lncRNA, miRNA, and mRNA (Figure 5b). According to the ranking results of PPI interaction networks and KEGG pathway enrichment results of DEPTGs, we selected six genes that may be related to IMF deposition: *HSP90AA1*, *PIK3CB*, insulin receptor subset 1 (*IRS1*), insulin receptor subset 2 (*IRS2*), lipin 1 (*LPIN1*), and perilipin 2 (*PLIN2*) (Figure 5c). These DEPTGs and some untargeted key genes regulate fat deposition through the mechanism shown in Figure 6.

## 4. Discussion

The quality of pork is an important economic characteristic that is significantly affected by IMF content. There are significant differences in meat quality among different breeds of pigs [37]. Compared with Western lean pigs, Chinese local pigs have higher fat deposition capacity. We compared mRNA, miRNA, and lncRNA in the LD of SL and Landrace pigs with different IMF contents to find the key factors affecting IMF deposition.

Previous studies have found many DEGs in the LD of obese and lean pigs through transcriptome sequencing technology [38,39,40]. In our study, 606 DEGs were found in the LD of both pig breeds, some of which are involved in glycogen synthesis and fatty acid metabolism, which may be responsible for the difference in IMF content between SL and Landrace pigs. AT-rich interaction domain 5B (*ARID5B*, also known as *MRF2*) is a member of the AT-rich interaction domain (ARID) family. The results showed that when *ARID5B* was knocked down in mouse embryonic fibroblasts and 3T3-L1 cells, adipogenesis was significantly inhibited [41]. Mice lacking *ARID5B* had reduced white fat mass and were resistant to obesity induced by a high-fat diet [42]. Carnitine palmitoyltransferase 1B (*CPT1B*) is the rate-limiting enzyme for β oxidation of long-chain fatty acids in the mitochondria of muscle cells. Increased expression of *CPT1B* in bovine fetal fibroblasts significantly increased triglyceride content [43]. Acyl CoA synthetase long-chain family member 1 (*ACSL1*) can participate in the initial step of fatty acid activation [44]. The overexpression of *ACSL1* will increase the triglyceride content in the liver [45]. In addition, the expression of *ACSL1* is significantly increased during the differentiation of preadipocytes under the pig skin [46]. The above three genes are highly expressed in the LD of SL pigs, so they may be the key genes affecting IMF deposition. In our study, *PLIN2* was considered to be another key gene affecting IMF deposition. It is highly expressed in the LD of SL pigs. *PLIN2* is a lipid droplet protein, which can be regarded as a protein marker of lipid droplets [47]. Lipids in lipid droplets can be catabolized by autophagy [48], but the overexpression of *PLIN2* will weaken autophagy and make lipid droplets accumulate. [49,50]. *PLIN2*, *CPT1B*, and *ACSL1* can participate in the PPAR signaling pathway. The peroxisome-proliferator-activated receptor (PPAR) is involved in regulating lipid metabolism. The up-regulation of the PPAR signaling pathway will be accompanied by the increase in lipid accumulation [51].

LncRNA is a key factor controlling gene expression [52]. LncRNA can regulate gene expression through cis and trans actions [53]. In addition, lncRNA can also be used as ceRNA to compete with mRNA to bind miRNA and then affect the expression of mRNA [54,55,56]. Our study showed that *LPIN1* is cis-regulated by MSTRG.13115.1 and MSTRG.13120.1, and trans-regulated by MSTRG.20210.1, MSTRG.10885.1, and MSTRG.19948.1. In addition, MSTRG.19948.1 can also act as a ceRNA and affect the expression of *LPIN1* by binding to ssc-mir-429. *LPIN1* is involved in the synthesis of triglycerides and phospholipids. The expression of *LPIN1* plays a key role in adipocyte differentiation, and it can also act as a nuclear transcriptional coactivator of some peroxisome-proliferator-activated receptors to regulate the expression of other genes related to lipid metabolism [57]. *LPIN1* was first found in fatty liver dystrophy mice, and the mutation of *LPIN1* can lead to fat malnutrition in fatty liver dystrophy mice, while overexpression of *LPIN1* can lead to obesity in mice [58]. The lack of *LPIN1* will lead to a significant reduction of adipose tissue quality and the abnormal expression of adipogenesis-related genes [59]. Wang et al. [60] showed that the expression of *LPIN1* in the LD of Rongchang pigs was higher than that of lean PIC pigs (PIC Swine Improvement Group, England, UK), and the expression of *LPIN1* in the *longissimus dorsi* muscle of Rongchang pigs with high IMF content was significantly higher than that of Rongchang pigs with low IMF content. Previous studies have shown that mir-429 can target the inhibition of LPIN1, affecting the PPAR signaling pathway [61]. Therefore, *PLIN1* may play an important positive regulatory role in the IMF sedimentation process. The LncRNAs MSTRG.19948.1, MSTRG.13120.1, MSTRG.20210.1, and miRNA ssc-mir-429 play an important role in regulating the expression of *LPIN1*.

Glucose and lipid metabolism is an important biological process for the body and cells to obtain energy and substances. Insulin and its signaling pathway are among the most important links in regulating glucose and lipid metabolism [62]. In our study, we also found that many DEPTGs can participate in the insulin signaling pathway, insulin resistance, PI3K-Akt-signaling pathway, and other related pathways. Insulin can promote glucose uptake and protein synthesis in muscle, and can promote fatty acid synthesis in adipose tissue and inhibit lipolysis [63]. The binding of insulin and insulin receptors will cause the aggregation of insulin substrate receptors. The latter can activate phosphatidylinositol 3-kinase (PI3K), and activate ser/thr kinases phosphoinositide dependent protein kinase-1 (PDK1) and protein kinase B (also known as Akt), thereby stimulating fatty acid synthesis and inhibiting gluconeogenesis [63,64]. *IRS1* and *IRS2* are similar in structure and function, but show tissue-specific differences. *IRS1* plays a leading role in skeletal muscle, while *IRS2* in skeletal muscle is negligible [65]. Studies have shown that the overexpression of *PGC1A* in primary hepatocytes can increase the expression of *IRS2* and decrease the expression of *IRS1*, and the high expression of *IRS2* may inhibit insulin-mediated gluconeogenesis [64]. *IRS1* can inhibit obesity induced by a high-fat diet through miR-503 [66]. Kovacs et al. [67] showed that the expression of *IRS1* in obese Indians was significantly lower than that in lean Indians. In our study, the expression of *IRS1* was also significantly lower in SL pigs than in Landrace. Studies have shown that miRNA-7 can effectively inhibit the expression of *IRS1* and *IRS2* [68]. *PIK3CA* and *PIK3CB* encode the catalytic subunit p110alpha and p110beta of PI3K, respectively [69,70]. *PIK3CA* and *PIK3CB* participate in the PI3K-Akt signaling pathway and mTOR signaling pathway, and the regulation of the PI3K-Akt-mTOR signaling pathway on lipid metabolism has been mentioned by many studies [71]. When the PI3K-Akt-mTOR pathway is inhibited, the intracellular lipid accumulation will be reduced, and the mRNA expression and protein content of genes related to de novo fatty acid synthesis will also be reduced [72]. Therefore, the high expression of *PIK3CA* and *PIK3CB* is conducive to the deposition of IMF. In addition, *IRS1*, *IRS2*, and *PIK3CB* are target genes for lncRNA MSTRG.10023.1, so MSTRG.10023.1 may play an important role in regulating the expression of the above three genes.

Heat shock protein (HSP) is a molecular chaperone which can reverse or inhibit the denaturation or unfolding of cellular proteins and play a protective role in cells [73]. HSPs are usually classified according to molecular weight and can be divided into HSP27, HSP40, HSP60, HSP70, HSP90, and large HSPs. In our study, heat shock protein family A (Hsp70) member 4 (*HSPA4*), heat shock protein family H (Hsp110) member (*HSPH1*), and *HSP90AA1* were at the core of the PPI network. Additionally, *HSP90AA1* is also affected by MSTRG.10023.1, MSTRG.10885.1, MSTRG.13120.1, MSTRG.19948.1, MSTRG.20210.1, MSTRG.4262.2, and ssc-mir-7-1. Studies have shown that overexpression of *HSPA4* leads to the activation of the Wnt signaling pathway [74]. The Wnt signaling pathway plays an important role in adipogenesis [75,76,77]. In addition, studies have shown that knocking down *HSPH1* and *HSP90AA1* can reduce cholesterol efflux [78]. The enrichment results of the KEGG pathway showed that *HSP90AA1* could participate in the PI3K-Akt signaling pathway. The PI3K-Akt signaling pathway is closely related to insulin resistance, and insulin resistance often leads to obesity [79,80]. Therefore, HSP90AA1 regulates adipogenesis by participating in the PI3K-Akt signaling pathway.

## 5. Conclusions

We constructed an interaction network of lncRNA, miRNA, and mRNA related to IMF, and through bioinformatics analysis, *ARID5B*, *CPT1B*, *ACSL1*, *LPIN1*, *HSP90AA1*, *IRS1*, *IRS2*, *PIK3CA*, *PIK3CB*, and *PLIN2* may be key genes related to IMF deposition. We found that several pairs of potential targeting relationships, MSTRG.19948.1 to *LPIN1*, *HSP90AA1*, *PIK3CB*, and MSTRG.19948.1 to ssc-miRNA-429 to *LPIN1*, MSTRG.13120.1 to *LPIN1*, *HSP90AA1*, *PIK3CB*, MSTRG.20210.1 to *LPIN1*, *HSP90AA1*, *PLIN2*, *IRS1*, MSTRG.10023.1 to *HSP90AA1*, *PIK3CB*, *IRS1*, *IRS2*, and ssc-miRNA-7-1 to *IRS1*, *IRS2*, may have an impact on IMF deposition. Our research can provide a reference for further understanding the regulatory mechanisms of IMF in the future.

## Figures and Tables

**Figure 1 genes-14-00168-f001:**
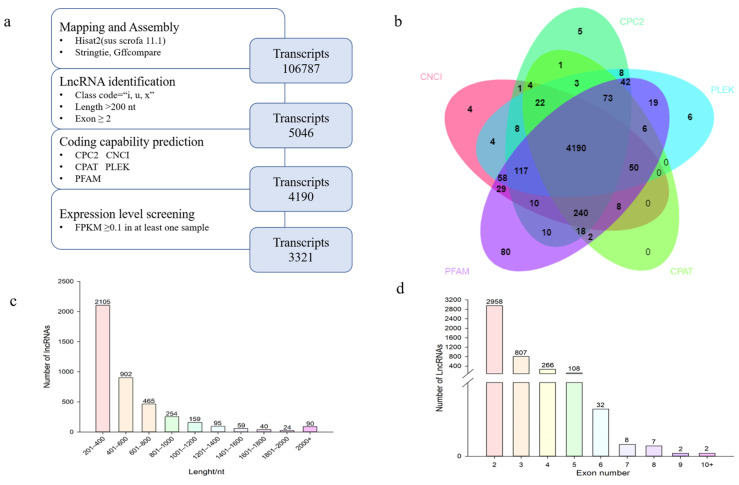
Identification and characterization of lncRNA. (**a**) Process of identifying lncRNA; (**b**) Venn plots of lncRNAs identified from Coding-Non-Coding-Index (CNCI), Coding Potential Calculator 2 (CPC2), predictor of long non-coding RNAs and messenger RNAs based on an improved k-mer scheme (PLEK), Coding Potential Assessing Tool (CPAT), and Pfam databases; (**c**) length distribution of lncRNAs; (**d**) exon number distribution of lncRNAs.

**Figure 2 genes-14-00168-f002:**
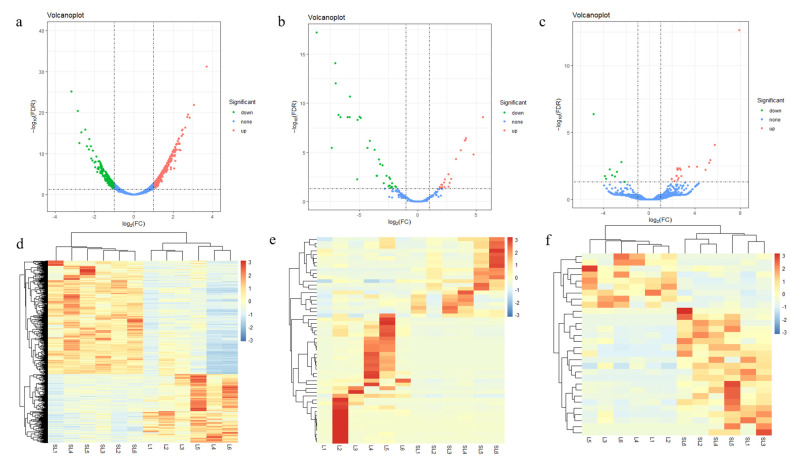
Three differentially expressed RNAs in Songliao black (SL) pigs and Landrace pigs. (**a**) Volcano plot of differentially expressed genes (DEGs); red represents up-regulation in SL pigs, green represents down-regulation in SL pigs, and blue represents no significant difference between SL pigs and Landrace, as below; (**b**) volcano plot of differentially expressed miRNA (DEMs); (**c**) volcano plot of differentially expressed lncRNAs (DELs); (**d**) heatmap of DEGs. The color code represents the expression value, red represents the high expression value, and blue represents the low expression value; SL stands for SL pig and L stands for Landrace; (**e**) heatmap of DEMs; (**f**) heatmap of DELs.

**Figure 3 genes-14-00168-f003:**
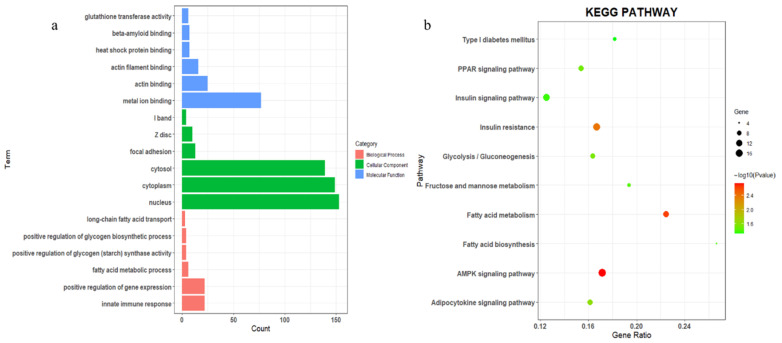
Functional enrichment analysis of differentially expressed genes (DEGs). (**a**) GO annotation of DEGs; (**b**) KEGG pathway analysis of DEGs.

**Figure 4 genes-14-00168-f004:**
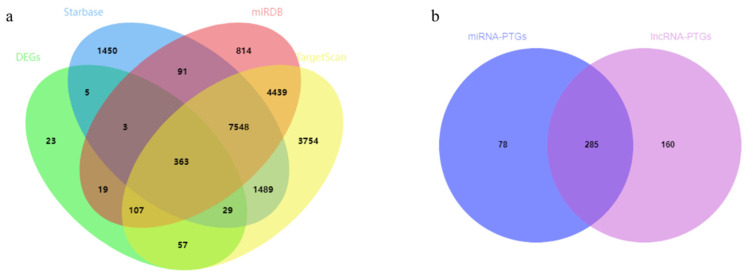
Venn diagram of target genes of differentially expressed miRNA and lncRNA. (**a**) Venn diagram of target gene prediction results of differentially expressed genes (DEGs) and three websites; (**b**) Venn diagram of target genes of differentially expressed miRNA (DEMs) and differentially expressed lncRNAs (DELs).

**Figure 5 genes-14-00168-f005:**
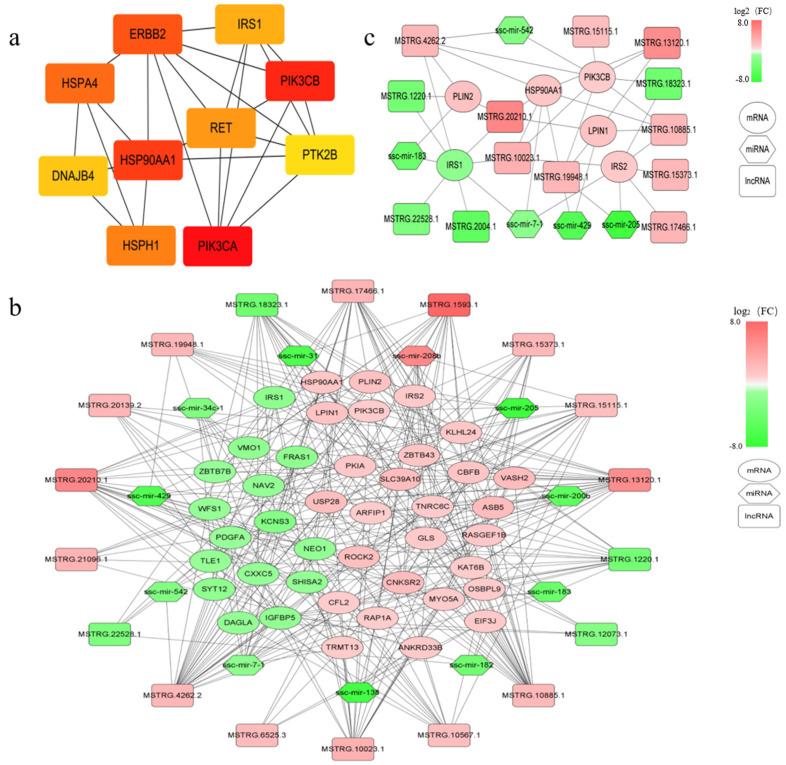
LncRNA–miRNA–mRNA network and protein–protein interaction (PPI) network hub gene. (**a**) The top 10 hub genes in the PPI network. The intensity of the color shows the connectivity ranking position: dark red indicates that the ranking is high, and light yellow indicates that the ranking is low; (**b**) interaction network diagram of lncRNA–miRNA–mRNA: red indicates up-regulation in Songliao black (SL) pigs and green represents down-regulation in SL pigs; (**c**) LncRNA–miRNA–mRNA interaction network of some key genes.

**Figure 6 genes-14-00168-f006:**
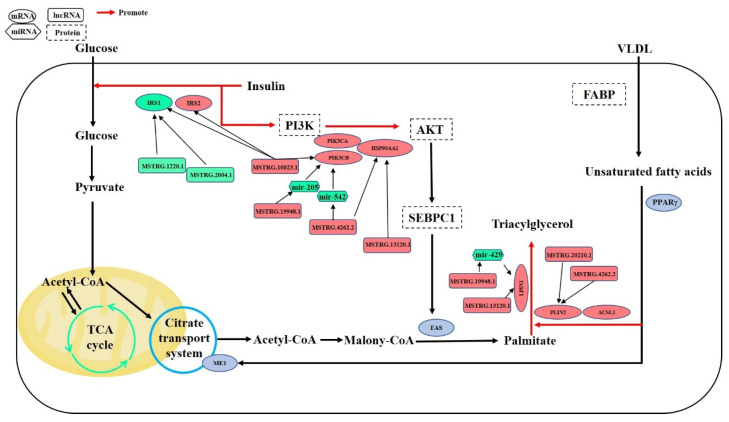
The major potential differentially expressed target genes (DEPTGs) and differentially expressed genes (DEGs) with lipid deposition in related pathways. Red indicates high expression in *longissimus dorsi* muscle (LD) of Songliao black (SL) pigs, and green indicates low expression.

**Table 1 genes-14-00168-t001:** The correlation between differentially expressed lncRNAs (DELs) and adjacent coding protein genes in differentially expressed genes (DEGs).

DEL	DEGs of Adjacent Protein Coding Genes	Distance (kb)	Spearman Correlation Coefficient	*p*-Value
MSTRG.10885.1	*DAPK3*	Coincide	0.812767	0.001311
MSTRG.13115.1	*LPIN1*	55	0.603267	0.037831
MSTRG.13120.1	*LPIN1*	33.2	0.802451	0.001683
MSTRG.17466.1	*DNAJB4*	41.8	0.777584	0.00291
	*FUBP1*	4.5	0.679511	0.01507
	*NEXN*	Coincide	0.735553	0.006402
MSTRG.21882.1	*CNKSR2*	Coincide	0.532917	0.074422
	*KLHL34*	45.7	0.691861	0.012674
MSTRG.383.1	*SYNCRIP*	29.2	0.313079	0.321749
MSTRG.6525.3	*HERC4*	13.8	0.751009	0.004874
MSTRG.8414.1	*DUSP1*	92.5	0.552691	0.062372

**Table 2 genes-14-00168-t002:** KEGG pathway enrichment analysis of potential differentially expressed target genes (DEPTGs).

Term	ID	*p*-Value	Gene
Insulin resistance	ssc04931	4.43 × 10^−7^	*RPS6KA1*, *CPT1B*, *PPP1R3C*, *IRS2*, *PIK3CA*, *PIK3CB*, *PRKAG2*, *PPARGC1A*, *PRKAA1*, *IRS1*, *SREBF1*, *PPARGC1B*, *G6PC3*
Insulin signaling pathway	ssc04910	1.96 × 10^−5^	*PPP1R3C*, *IRS2*, *PIK3CA*, *PIK3CB*, *PRKAG2*, *PPARGC1A*, *PRKAA1*, *IRS1*, *SORBS1*, *SREBF1*, *G6PC3*, *EIF4E*
Adipocytokine signaling pathway	ssc04920	0.000106564	*CPT1B*, *IRS2*, *ACSL1*, *PRKAG2*, *PPARGC1A*, *PRKAA1*, *IRS1*, *G6PC3*
mTOR signaling pathway	ssc04150	0.000821208	*RPS6KA1*, *DDIT4*, *PIK3CA*, *PIK3CB*, *IRS1*, *PRKAA1*, *CLIP1*, *EIF4E*, *WNT5B*, *RICTOR*
PI3K-Akt signaling pathway	ssc04151	0.000919504	*ERBB2*, *PDGFA*, *DDIT4*, *IL4R*, *PPP2R3C*, *PIK3CB*, *IRS1*, *PRKAA1*, *PIK3CA*, *VEGFA*, *ITGAV*, *COL1A1*, *EIF4E*, *HSP90AA1*, *MYC*, *G6PC3*
Non-alcoholic fatty liver disease (NAFLD)	ssc04932	0.002901411	*DDIT3*, *PIK3CA*, *EIF2AK3*, *PRKAG2*, *IRS1*, *PRKAA1*, *SREBF1*, *IRS2*, *PIK3CB*
Regulation of lipolysis in adipocytes	ssc04923	0.005012828	*PIK3CA*, *PIK3CB*, *IRS1*, *IRS2*, *ABHD5*
Glycerolipid metabolism	ssc00561	0.006591252	*GPAM*, *LPIN1*, *LCLAT1*, *DGKD*, *DGKE*
PPAR signaling pathway	ssc03320	0.016374973	*SORBS1*, *ACSL1*, *PLIN2*, *ACADM*, *CPT1B*
Fatty acid metabolism	ssc01212	0.026319948	*FADS1*, *ACSL1*, *ACADM*, *CPT1B*

## Data Availability

The data presented in this study are available upon request from the corresponding author.

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
