# Peer review of "Comprehensive Analysis of the lncRNA–miRNA–mRNA Regulatory Network for Intramuscular Fat in Pigs"

_genes, 2023, doi:10.3390/genes14010168_

Round 1

Reviewer 1 Report

Intramuscular fat (IMF) is an important index for meat quality. This study by Zhao and his co-authos focus on IMF trait in Songliao Black pigs and Landrace pigs using RNA sequencing technology. Several genes, miRNA and lncRNAs have been identified by network analysis. The results provide new insight into the genetic mechanism of IMF, however, there are still several comments should be taken into consideration:

1.      The current version of manuscript has much words spell problems, the authos should be check carefully.

2.      The conclusion was pretty simple. Only genes apper to play important role in IMF from your conclusion. In fact, this study investigated the lncRNA-miRNA-mRNA interaction network that influence the IMF deposition. The key miRNAs, lncRNAs or pathways should be the highlights of this study besides genes.

3.      The term how selecte? Top or according the authors knowlogy? Were all GO terms and KEGG pathwas list in Figure3? If not, How were the critertia? Top 6 for GO and top 20 for KEGG, or based on your experience? In figure3, the full names of BP, CC and MF should be given in the legend.

4.      What did the figure 2d, e and f shown? There were no description about the results of these heatmaps.

5.      Line18, 41, 53,148, 219: ‘landrace’ change to ‘Landrace’,

6.      Line18: Most abbreviations used in this study should be expanded at first use, such as DEG, DEM, DEL.

7.      Line20: ’a’ to ‘an’

8.      Line91: It would be better that the citation position of Hmmer3 tool is closely followed by the softare, because this centance much too long.

9.      Line153: ‘go’ to ‘GO’

10.  Line230: ‘carnitine’ to ‘Carnitine’

11.  Line247: ‘CIS’ to ‘cis’

12.  Line260: ‘Showen’ to ‘showed’

13.  Line261: ‘rongchang’ to ‘Rongchang’

14.  Line269: ‘HSP 90’ to ‘HSP90’

15.  Line270: ‘(Hsp70)’ should be deleted?

Reviewer 2 Report

In the study by Zhao et al. a differentially expression analysis of mRNA, lncRNA and miRNA in longissimus dorsi muscle between two extreme pig breeds for intramuscular fat (IMF) content was performed to determine the regulatory mechanisms of IMF. Although the integration of the different types of RNAs is an interesting approach to deepen in the regulatory mechanism that affects IMF content, the present article presents some weaknesses:

(1)    The comparison is made between two extreme breeds for fat deposition. Landrace pig selected as a lean pig while Songliao Black pig with good fat deposition ability. Genetic architecture of muscle will be very different between animals, not only at the adipocyte level but also at the muscle cell level. In addition, the determination of the phenotype to be analyzed and the differences between the sequenced animals (percentage of intramuscular fat) are not presented. A better experimental design would have been to select animals within the Songliao Black pig with extreme percentages of IMF.

 (2)    The sequencing depth is low (average of 10.66 million of reads per sample). The ENCODE Consortium suggests ~30 million raw reads per sample (30 M PE reads of length> 30NT, of which 20-25M are mappable to the genome or known transcriptome) in experiments whose purpose is to evaluate the similarity/differences between the transcriptional profiles of two polyA+ samples.

 (3)    The material and methods section don’t provide sufficient information. In section 2.2 is missing the kit for preparing the libraries, the length of the reads, the depth of sequencing… In section 2.3 is missing the information about the version of the annotation database. Section 2.7: explain better how the prediction of target genes for lncRNA in trans was made and how was constructed the interaction network using Cytoscape. Also the reference of the Cytoscape software is missing.

(4)    In the results section provide an ensembl id for the lncRNA and miRNA of tables S3 and S4. It is very difficult to extrapolate results according to the nomenclature assigned to the two types of RNAs. Also, the reviewer did not find in the results section the information of targeted miRNA of DELs and intersected with DEMs. Regarding section 3.4, lines 154-157: where are these terms and pathways in the Figure 3a?

In section 3.5, line 166: provide a supplementary table with the 363 genes and their predicted miRNA. Add the distance (kb) between the lncRNA and their predicted target gene in Table 1. In line 178, the number of 285 PTGs is not appearing in Figure 4b. Section 3.6, lines 197-200: justify better the selection of the six genes to construct the network.

 (5)    Discussion: The discussion is very speculative and is only focused on a limited number of DEGs, and the direction of the expression regarding these results is not discussed (only for the PLIN2 gene). Also, discussion about DEGs regulated by miRNAs is missing. Line 252: is the LPIN1 gene and ssc-mir-429 associated in the results, and the ssc-mir-429 and the MSTRG.19948.1? Explain better the hypothesis based on the results regarding the MSTRG.19948.1, the LPIN1 and the ssc-mir-429 RNAs?

Lines 282-310 I suggest moving this section at the beginning of the discussion.

(6)    The color information is missing in the Figure 2; up and down genes by which comparison? Add in the figure 3 caption the gene ontology categories. The size of the legends is very small. Figure 5: what does it mean ranking position? The color information is also missing

 Minor comments:

I suggest authors to avoid expressions such as: “attracted our attention”, “we believe”, “we thought”

Abstract: add the experimental approach, line 19: two groups of what?

Write gene names in italics throughout the manuscript.

Line 198: six genes instead of seven genes?

Reviewer 3 Report

Line 15: “(…), and it is regulated by multiple genes…”

Line 17: Longissimus dorsi in italics. Repeat the same correction in the rest of the manuscript.

Line 22: ARID5B, CPT1B, ACSL1, LPIN1, HSP90AA1, IRS1, IRS2, PIK3CA, PIK3CB, PLIN2. Name of genes in italics

Line 28: main source

Line 28: space after the end of sentence

Line 49: greater fat deposition

Line 51-53: reword sentence

Line 53-54: I do not believe that this sentence is necessary. This sentence should be your objectives more than your “result”. If that is the case, I suggest to reword the sentence.

Line 57-61 Materials and methods:

-        Further explanations about the diet should be explained. The diet received plays a critical role in the IMF deposition. Furthermore, the diet could impact differently depending on the breed of the animal receiving the diet. In this study, two different breeds were tested; therefore showing the characteristics of the diet is critical.

-        Which was the feed intake of each animal per day?

-        Also it is necessary to report the initial body weight and the final body weight of the animals.

-        Where were the pigs raised? Were there in confinement? What was the location of the experiment? Weather condition is important to mention, in at least one or two sentences, because weather has an impact on gene expression, even more profound when analyzing different breeds.

Line 57: Songliao Black (n = 6) and Landrace pigs (n = 6)

Line 57: Are all the animals male?

Line 58: Approximately 100kg

Line 59-60: Add reference.

Line 61: what was the treatment? Treatment or analysis?

Line 66: specify where bio rad is located in USA.

Line 66: how did you detect the integrity? What criteria did you follow for measuring that integrity? Did you have access to measure the integrity of DNA with NanoDrop?

Line 77: When describing a process, use “First”, “Second”, etc. I would change the phrase “after that”.

Line 91: Same comment than Line 77

Line 130: “We detected that the comparison rate of all samples (…)”

Line 138: Add figure title. Improve the quality of graphs and figure. Ensure the legibility of the names in figures. In figure 1b, make sure that all the names are aligned in x axis.

Line 153: GO instead of go.

Line 154-157: Explain why you were interested in those terms and then re-word that sentence.

Line 161: Figure 3 in bold. Make sure to maintain consistency in writing “Figure X” in bold.

Line 162: Explain the abbreviation BP, CC, MF in the figure title. The figure should be self-explanatory.

Line 163: Subtitle in italics. Make sure that the rest of the subtitles are in italics as well.

Line 173-174: Convert those two sentences in one.

Line 175: “differs” instead of is very different.

Line 185: Figure 4 in bold. Make sure that all letters in the figure are clear and readable.

Line 188-189: Table 1 in bold. All gene names have to be written in italics.

Line 191-202: Gene names in italic.

Line 214: words “important” and “pork quality” repeated. Reword to improve readability.

Line 215: There is

Line 216-218: Reword that argument. The fact that IMF is complex and regulated my multiple genes does not contradict that Chinese local pigs have higher fat deposition.

Line 218-220: I am sorry, I did not find where the IMF content between breeds was reported. Did you arrive to that argument based on literature? Did you have the possibility to report fat content, carcass yield and overall meat quality of the two breeds after the slaughter? I recommend expanding more the results obtained during feeding period and carcass quality in order to have a better background for discussing the gene expression results.

Line 226: Also known as

Line 227-230: What happened with ARID5B? Was it downregulated or upregulated in the present study? I suggest to start mentioning what happened in the study and then utilize the reference to support that result.

Line 230: Carnitine.

Line 236: Avoid starting sentences with abbreviations.

Line 241: Reword references.

Line 251: In addition instead of not only that

Line 256: Do not use abbreviation if you are not using it again in the manuscript.

Line 261: What is PIC pigs? That abbreviation was never used before in the manuscript

Line 265: Good argument. Did those studies in the references showed the interaction with PPARg? I consider that this argument is important since it is widely recognized that PPARg is the key regulator of adipogenesis.

Line 282-284: Add references

Line 283: What do you mean by material sources?

Line 283: Insulin and its signaling pathway (…)

Round 2

Reviewer 2 Report

1. Please check the tenses of the modified paragraph in section 2.1.

2. The sequencing depth is low. The authors have to indicate it as a drawback of their study although other studies have reported the same sequencing depth.

In table S1. The total  number of L6 reads seems not to be correct compared to the other samples.

3. In section 2.2 the following information is still missing: the kit for preparing the libraries, the length of the reads, the depth of sequencing,..

In section 2.3 the version of the genome annotation file is still missing

About my comment to better explain how the authors built the interaction network is still not clear to me. Were correlation values between all RNA types included? Explain better.

4. In the results section and tables S3 and S4, the available information of ensembl id codes for lncRNA and miRNA are still missing.

In line 165 of the clean version: change fat acid metallic process for fatty acid metabolic process

Regarding the justification of the six genes to construct the network: line 210: avoid the use of "we think" and add bibliography to support the statement

5. Discussion: line 247, delete "the" and add PLIN2 in italics

line 275: modify the sentence as follows: this gene may be a key gene...

line 277: LPIN1 in italics.

Other comments:

Lines 151-153: in my opinion the heatmap of miRNAs is not differentiating the groups very well , modify the sentence.

Reviewer 3 Report

I strongly reccomend to include information regarding initial body weight and age of animals at the beggining of the study. Also, it is important to show the diet (ingredients and proportions of each ingredient).
